# STRUCTURE-AWARE ANTIBODY HUMANIZATION VIA A NOVEL BLACK-BOX OPTIMIZATION ALGORITHM

## ABSTRACT

Antibodies are essential for viral neutralization and therapeutic applications. However, non-human antibodies can provoke anti-antibody immune responses, prompting the development of humanization strategies to reduce immunogenicity. Most existing approaches focus primarily on improving humanness scores from a sequence perspective, often overlooking the structural stability of humanized antibodies, which is essential for preserving complementarity-determining region (CDR) conformations. To overcome this limitation, we propose Hu-MCTs, a two-stage framework comprising (1) a pretraining phase that learns latent representations of human antibody sequences, and (2) a humanization phase that optimizes murine sequences toward human-like sequences using a novel black-box optimization algorithm based on **M**onte **C**arlo **T**ree **S**earch. This algorithm jointly considers humanness and structural integrity, particularly minimizing disruption to CDR conformations. Experimental results demonstrate that Hu-MCTs outperforms baseline methods by achieving higher humanness scores while better preserving CDR structural stability. Moreover, the generated sequences exhibit the highest biological plausibility scores, closely resembling natural antibodies. These results suggest that Hu-MCTs is an effective solution for humanizing antibodies while preserving key structural features for functionality.

## 1 INTRODUCTION

Antibodies, or immunoglobulins (Igs), are Y-shaped macromolecules composed of two identical heavy chains and two identical light chains, connected by disulfide bonds (Dondelinger et al., 2018). Each chain's variable region contains three hypervariable segments known as complementarity-determining regions (CDRs). Together, the six CDRs form the antigen-binding site, or paratope, which interacts with a specific epitope on an antigen, enabling high-affinity binding, as shown in Figure 1(a) (Davies & Chacko, 1993). This specificity makes antibodies central to immune defense and highly successful as targeted therapeutics (JJ, 2000).

Murine antibodies are pivotal in early drug discovery due to their rapid and cost-effective production via hybridoma technology (Lu et al., 2020). However, their non-human origin poses a significant risk of eliciting human anti-antibody responses (AAR), which can compromise efficacy and safety (Tjandra et al., 1990). As illustrated in Figure 1(b), immunogenicity risk is directly proportional to the non-human content of an antibody. Consequently, **humanization**—engineering non-human antibodies to resemble human sequences—is a critical step in their clinical development (Fransson, 2008). The conventional approach, CDR grafting, transfers murine CDRs onto a selected human framework region (FR) (Lo, 2004). While effective at reducing immunogenicity, this process can disrupt the precise CDR conformations essential for antigen binding, often leading to a loss of affinity and requiring laborious, intuition-driven back-mutations to restore function (Safdari et al., 2013a).

To overcome these limitations, recent computational methods have leveraged protein language models (PLMs). Approaches like Sapiens (Prihoda et al., 2022), Hu-mAb (Marks et al., 2021a), and Humatch (Chinery et al., 2024) have improved humanness from a sequence perspective. However, they often neglect the critical role of **structural integrity**, failing to explicitly optimize for the preservation of CDR loop conformations necessary for maintaining binding affinity. While structure-aware

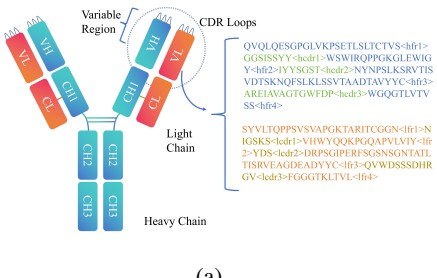 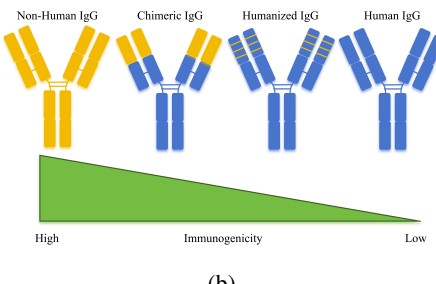

(a)                                                    (b)

Figure 1: **The structure of antibody and Correlation between humanization degree and immunogenicity reduction.**(a) The antibody consists of two heavy chains and two light chains. Within the variable region, the CDR loops interact directly with antigens (especially CDR3), enabling the antibody to identify and bind to target molecules precisely.(b) This progression demonstrates that as the degree of humanization increases-from non-human to fully human sequences-the immunogenicity of the antibodies diminishes accordingly.

methods like CUMAb (Tennenhouse et al., 2022) exist, their exhaustive candidate screening and structure prediction pipelines are computationally prohibitive for rapid design cycles.

The field requires a method that efficiently navigates the vast sequence space to co-optimize sequence humanness and structural function. To address this gap, we introduce **Hu-MCTs** (Humanization via Monte Carlo Tree Search), a novel black-box optimization framework. While its core components—latent space sampling and MCTS (Yang et al., 2022; Wang et al., 2022)—have been used in broader molecular design, our work is, to our knowledge, the **first to apply and tailor this paradigm to the unique constraints of antibody humanization**. Our key innovations are:

1. **Joint VH/VL Latent Space Exploration:** We introduce a Conditional Variational Autoencoder (CVAE) to model the joint latent space of paired heavy (VH) and light (VL) chains. This is a critical advance, as it allows for simultaneous humanization that accounts for inter-chain dependencies essential for structural stability and function.

2. **Multi-Objective, Structure-Guided Optimization:** Our framework employs a multi-objective MCTS algorithm, seeded by a novel PLM-guided strategy. This allows for a balanced search that jointly optimizes for high humanness scores and, crucially, **preserves structural integrity**. We are the first to incorporate the preservation of parental CDR conformation, measured by RMSD, as a direct objective in the humanization process, addressing a key limitation of prior work.

Through systematic benchmarking, we demonstrate that Hu-MCTs outperforms methods in achieving superior humanness while maintaining high structural fidelity. By eliminating template dependence and integrating multi-objective optimization of sequence and structure, our approach provides a robust and efficient solution for developing safer and more effective antibody therapeutics.

## 2 RELATED WORK

**Antibody humanization** Recent advancements in antibody humanization have led to several new approaches. Hu-mAb (Marks et al., 2021b) and MG (Clavero-Álvarez et al., 2018) use a humanness score to guide the process, while Sapiens (Prihoda et al., 2022), a sequence-based method, employs two transformer models to humanize the framework region without relying on scoring. The CUMAb (Tennenhouse et al., 2024) method incorporates structural information by building 3D antibody models, inserting mouse CDRs, and selecting the lowest-energy structure. Humatch (Chinery et al., 2024) offers fast humanization of both heavy and light chains using three Convolutional Neural Networks (CNNs), aligning sequences with experimental data to improve stability and reduce immunogenicity. These methods differ significantly from traditional approaches like CDR grafting (Hu et al., 2014; Safdari et al., 2013b), which focus mainly on framework region sequence similarity.

**Protein language model** Models like the ESM series (Rives et al., 2021; Meier et al., 2021; Lin et al., 2023; Hayes et al., 2025), especially ESM-3, have shown strong performance, such as in discovering new fluorescent proteins. CARP (Yang et al., 2024), which uses a CNN instead of a transformer, also performs well. In addition to general protein models, many methods focus specifically on antibody sequences for tasks like sequence recovery, expression prediction, and binding affinity estimation. IgBERT and IgT5 (Kenlay et al., 2024) are pretrained on unpaired sequences and fine-tuned on paired antibody data. BALM (Jing et al., 2024) introduces a special architecture called Baformer to better capture antibody-specific features. AbLang2 (Olsen et al., 2024) reduces germline bias by predicting non-germline residues. AntiBERTa (Leem et al., 2022) predicts paratope positions, and AntiBERTy (Ruffolo et al., 2021) applies a Multiple Instance Learning approach to identify antigen-binding sites. IgLM (Shuai et al., 2021) improves antibody design using large datasets and bidirectional context. pAbT5 (Chu & Wei, 2023) generates one antibody chain from its paired partner, modeling key properties like CDR diversity and chain pairing.

## 3 METHOD

This section describes the Hu-MCTs framework, which integrates pre-training of core models and a humanization pipeline to achieve template-free antibody optimization. The method is structured into two sequential phases: a pre-training phase to learn latent representations of human antibodies, and a humanization phase to iteratively optimize murine sequences toward human-compatible variants while preserving CDR structure and antigen binding specificity, as illustrated in Figure 2.

### 3.1 MODEL PRE-TRAINING

This study develops two core models: a Protein Language Model (PLM) for selecting humanization initial points and a Conditional Variational Auto-Encoder (CVAE)-based encoder-decoder model for antibody encoding and generation.

The PLM is built by fine-tuning the IgBERT model (Kenlay et al., 2024; Elnaggar et al., 2021), with the aim of capturing residue-residue and residue-region relationships in paired antibodies. During training, each residue is augmented with a 17-dimensional one-hot label indicating its regional identity (e.g., CDR1-3, FR1-4 for heavy/light chains, and special tokens like CLS/SEP/PAD). The model's embedding layer is modified to fuse residue token embeddings with region label embeddings via a Multi-Layer Perceptron (MLP). Training follows the Masked Language Model (MLM) objective (Devlin et al., 2019), where 15% of residues are randomly masked, and the model learns to reconstruct them, enabling context-aware sequence representations.

For the CVAE-based encoder-decoder model, we adopt the IgT5 encoder-decoder architecture (Kenlay et al., 2024; Elnaggar et al., 2021) and trained on our antibody dataset. To enhance latent variable utility for humanization, the token table is expanded with region boundary markers (e.g., indicating CDR/FR start/end) and a sequence-level marker $</z>$ at the input start.

Let $x$ denote the Framework (FR) regions and $c$ denote the conditioning CDR regions. The encoder, which receives both $x$ and $c$, generates the parameters $\mu$ and $\sigma^2$ for the approximate posterior distribution $q_\phi(z|x,c)$. A latent variable $z$ is then sampled using the reparameterization trick. The decoder reconstructs the FR regions from $z$ and the conditioning CDR information $c$.

Training optimizes the conditional Evidence Lower Bound (ELBO) (Kingma & Welling, 2022):

$$\mathcal{L}(x,c) = \mathbb{E}_{q_\phi(z|x,c)}\left[\log p_\theta(x|z,c)\right] - \mathcal{D}_{KL}\left(q_\phi(z|x,c) \parallel p(z)\right) \tag{1}$$

where the first term is the reconstruction loss for the FR regions, conditioned on both the latent variable $z$ and the CDRs $c$. The second term is the KL divergence, which regularizes the approximate posterior $q_\phi(z|x,c)$ to be close to a standard Gaussian prior $p(z)$. To mitigate posterior collapse, cost annealing (Bowman et al., 2016) is employed, where a $\beta$ coefficient gradually increases during training to upweight the KL divergence term.

---

**Algorithm 1** Hu-MCTs Pseudocode for Antibody Humanization

---

**Require:**
1: Number of rounds $T$, Evaluation Function $f(\mathbf{x})$ (combining humanization score and structural stability), Human Antibody Dataset $\mathcal{D}$, PLM Model $p(\cdot)$, CVAE Model $c(\cdot)$, Initial number of samples $N_{\text{init}}$, Re-partitioning interval $N_{\text{par}}$, Node partition threshold $N_{\text{thres}}$, UCB parameter $C_p$

**Ensure:** Optimized humanized antibody sequence
2: Pre-train $p(\cdot)$ on $\mathcal{D}$ for humanness scoring; Pre-train $c(\cdot)$ on $\mathcal{D}$ for latent space encoding/decoding.
3: Set region partition $\mathcal{V}_0 = \{\Omega\}$
4: Draw $N_{\text{init}}$ samples $\{\mathbf{x}_i\}_{i=1}^{N_{\text{init}}}$ from $\mathcal{D}$ using $p(\cdot)$ (selecting most similar to murine antibody).
5: **for** $t = 0, \ldots, T - N_{\text{init}} - 1$ **do**
6:     **if** $t \bmod N_{\text{par}} = 0$ **then**
7:         Re-learn region partition:
8:         $\mathcal{V}_t \leftarrow \text{Partition}(\Omega, \mathcal{S}_t, N_{\text{thres}}, p(\cdot), c(\cdot))$ in latent space of $c(\cdot)$
9:     **end if**
10:     **for** $k := \text{root}, k \notin \mathcal{V}_{\text{leaf}}$ **do**
11:         $k \leftarrow \arg\max_{\Omega_c \in \text{child}(\Omega_k)} b_c$ where
12:         $b_c := \max_{\mathbf{x}_i \in \Omega_c} f(\mathbf{x}_i) + C_p \sqrt{\frac{2 \log n(\Omega_k)}{n(\Omega_c)}}$
13:     **end for**
14:     Initialize CMA-ES using encodings of $\mathcal{S}_t \cap \Omega_k$ via $c(\cdot)$ , $\Omega_k$ is the chosen leaf sub-region.
15:     $\mathcal{S}_t \leftarrow \mathcal{S}_{t-1} \cup \{(\mathbf{x}_t, f(\mathbf{x}_t))\}$ , $\mathbf{x}_t$ is drawn from CMA-ES and decoded via $c^{-1}(\cdot)$.
16: **end for**

---

## 3.2 HUMANIZATION

Our humanization pipeline integrates pretrained models and black-box optimization (Reidenbach, 2024) to generate humanized antibody sequences, balancing human-like properties and structural stability. The workflow proceeds as follows:

First, we use the trained Protein Language Model (PLM) to select initial candidates. For a murine antibody input, the PLM generates its latent representation—capturing residue-level features and region-specific context learned via MLM training. We then compute cosine similarity between this latent vector and all human antibody sequences in our preprocessed OAS database (1,738,321 paired sequences). To ensure the initial templates provide a diverse set of starting points, we select $K$ candidates using a Maximal Marginal Relevance (MMR) approach. This method balances the candidates' similarity to the murine sequence with their dissimilarity to each other, thus preventing the selection of a redundant, homogenous set. The impact of this hyperparameter $K$ on model performance is further analyzed in Appendix D. This step anchors the humanization process in a varied and biologically plausible set of human frameworks.

Next, the $K$ human candidates and murine input are processed through the CVAE encoder-decoder model. The murine antibody's CDR regions are encoded into a condition vector $\mathbf{c}$ (preserving CDR structural/functional information critical for antigen binding). Each human candidate is independently encoded by the CVAE encoder, yielding latent variables $z_1, z_2, \ldots, z_K \sim \mathcal{N}(\mu, \sigma^2)$ via the reparameterization trick. Concatenating $z_i$ with $\mathbf{c}$ forms $\mathbf{z}_{\text{cond}} = [z_i; \mathbf{c}]$, which the decoder uses to generate humanized sequences—preserving murine CDRs while humanizing framework regions.

To optimize the latent space search, we adopt Monte Carlo Tree Search (MCTS) (Yang et al., 2022; Wang et al., 2022), a black-box optimizer well-suited for non-differentiable, discontinuous objective functions (see Algorithm 1). The implementation of the space-partitioning function and CMA-ES is provided in Appendix H. The MCTS tree represents the search space, with nodes corresponding to subregions that contain latent variables $z$. Starting from the root node, the UCB formula guides node selection:

$$\text{UCB}(n) = \frac{Q(n)}{N(n)} + \sqrt{\frac{2 \ln N(\text{parent})}{N(n)}} \tag{2}$$

where $Q(n)$ is node $n$'s cumulative reward, $N(n)$ its visit count, and $N(\text{parent})$ the parent's visits. This balances exploration (under-visited nodes) and exploitation (high-reward nodes).

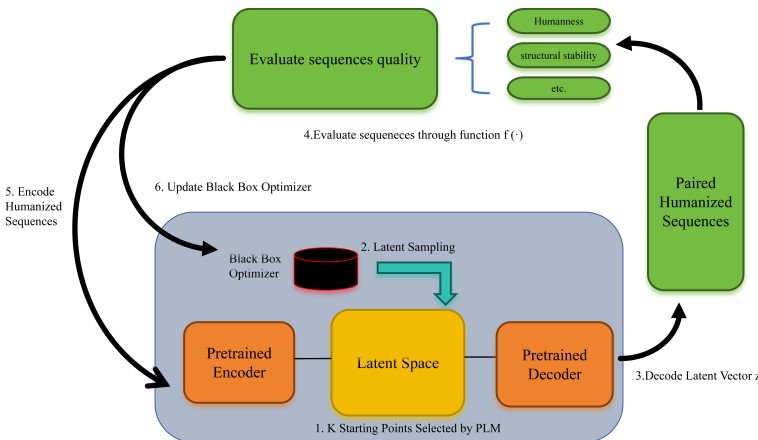

Figure 2: **Humanization pipeline.** The pipeline commences with selecting K starting points via a PLM. The sampled latent vectors are decoded by a pretrained decoder to yield paired humanized sequences. These sequences are then evaluated through function $f(\cdot)$ for attributes such as humanness and structural stability. Following this evaluation, the humanized sequences are re-encoded, and the black-box optimizer is updated using this feedback.

Each candidate $z$ is decoded into a sequence and evaluated using two objectives:

1. **Humanization Score ($S_{\textbf{human}}$)**: Computed as the maximum Abnativ score (Ramon et al., 2023) for heavy and light chains of the paired antibody.

2. **Structural Stability ($S_{\textbf{stab}}$)**: Derived from Tfold-predicted 3D structures (Wu et al., 2022), measuring CDR structural preservation via RMSD, weighted by CDR importance ($w_{\text{CDR3}} > w_{\text{CDR2}} > w_{\text{CDR1}}$):

$$S_{\text{stab}} = \sum_{i=1}^{3} \sum_{X \in \{H,L\}} w_i \cdot \text{RMSD}\left(\text{CDR}_{\text{human},i,X}, \text{CDR}_{\text{murine},i,X}\right)$$

The combined reward for MCTS is defined as $\text{Reward}(z) = \alpha \cdot S_{\text{human}} - \beta \cdot S_{\text{stab}}$, prioritizing humanization while penalizing structural deviation. Reward values propagate up the tree to update node priorities.

Finally, MCTS-identified promising latent regions undergo local optimization via the Covariance Matrix Adaptation Evolution Strategy (CMA-ES) (Hansen, 2023), which is more efficient in high-dimensional spaces ($d > 100$) than TuRBO (Eriksson et al., 2020). CMA-ES samples $z \sim \mathcal{N}(\mu, \Sigma)$ within these regions, iteratively refining $\mu$ and $\Sigma$ to maximize $\text{Reward}(z)$. This refines MCTS results, ensuring convergence to high-quality humanized sequences.

In summary, our pipeline combines PLM-guided initialization, CVAE-based generation, MCTS global search, and CMA-ES local optimization to balance human-like sequence properties and structural stability.

## 4 EXPERIMENTS

### 4.1 SETUP

#### 4.1.1 DATASETS

**Training Set**: 1.7 million paired human IgG/IgA sequences with IMGT-numbered (Lefranc et al., 2015) CDRs from OAS, used for CVAE and PLM training. **Test Set**: Humab25 dataset comprises 25 experimentally validated paired antibodies derived from the Hu-mAb. Additionally, we compile HuAb348 dataset from patent records, which comprises 348 experimentally validated humanized paired antibodies and their corresponding parental mouse antibodies. These datasets serve as test

sets to evaluate our model's performance against Sapiens and traditional methods, as discussed in the main text.

### 4.1.2 BASELINE METHODS

We compare against five approaches, with key characteristics summarized from related work:

1. **Experiment**: Experimentally validated humanized antibody sequences from wet-lab studies. 2. **Sapiens** (Prihoda et al., 2022) : A transformer-based tool in the BioPhi platform, trained on OAS via masked language modeling. It prioritizes 9-mer peptide frequency (OASis score) but processes heavy/light chains independently, risking inter-chain instability. 3. **Hu-mAb** (Marks et al., 2021a) : Employs human gene-specific random forest classifiers to guide iterative single-point mutations. While accurate in V-gene identification, its chain-independent processing limits CDR flexibility and may disrupt inter-chain stability. 4. **Humatch** (Chinery et al., 2024) : Uses lightweight CNNs to jointly humanize heavy/light chains, prioritizing VH/VL compatibility. Though fast (sub-second processing), it focuses on germline guidance and may under-explore non-homologous frameworks. 5. **CUMAb** (Tennenhouse et al., 2022) : A structural approach grafting CDRs onto thousands of human frameworks, validated via Rosetta energy minimization. It excels in structural preservation but requires atomistic simulations, limiting throughput.

### 4.1.3 EVALUATION METRICS

**Humanness**: $S_{\text{humanness}}$: For benchmarking, we select the widely recognized Sapiens model and use its evaluation metrics (OASis and T20 scores) to assess improvements in humanness.Additionally, we introduce a metric called Germline Identity, which measures the similarity between the framework region (FR) sequences of the humanized antibodies and those of the closest human germline. **Structure**: RMSD$_{\text{CDR}}$: we calculate all-atom RMSD and backbone ( $C\alpha$, $C$, $N$, $O$) atom RMSD (Å) of the three CDR regions . **Pairing Quality**: Humatch tool-based CNN-P scoring for antibody pairing quality assessment. **Sequence Alignment**: In the evaluation of the similarity between humanized antibodies and their experimental counterparts, we focus on two key metrics: Preservation: This metric measures the extent to which residues from the original mouse sequences are retained in the humanized antibodies. Specifically, "Total" refers to all residues in the sequence, while "Vernier" denotes the key residues defined by the Kabat numbering scheme. Achieving closer alignment with experimental results is preferable for this metric. Mutation precision: This assesses the consistency of mutated residues between the humanized antibodies and experimental results.

### 4.2 RESULTS

Our experimental results demonstrate the effectiveness of our humanization pipeline across key metrics, with detailed numerical comparisons provided in Tables.

### 4.2.1 SEQUENCE HUMANNESS AND PRESERVATION.

Our model, Hu-MCTs, significantly surpasses all baseline methods in key humanness metrics. As detailed in Table 1, it achieves the highest OASis score (41.63%), T20 scores for both heavy and light chains, and nearly perfect germline identity (98.77% and 99.64%, respectively). This performance stands in sharp contrast to conservative methods like Hu-mAb, which barely improves upon the parental murine sequence (0.074% OASis). This advantage stems from our framework's synergistic design: the pre-trained language model (PLM), conditional variational autoencoder (CVAE), and MCTS algorithm effectively guide the search toward natural and diverse human sequence patterns.

A key aspect of our strategy is a more comprehensive modification of the murine sequence, which is reflected in the mutation counts. Hu-MCTs introduces a higher number of mutations (33.88 in the heavy chain, 27.92 in the light) compared to methods that prioritize minimal changes, such as Hu-mAb (13.72 and 8.20). As explained in Appendix C.1, this is a deliberate strategic choice to more effectively eliminate murine epitopes and reduce immunogenicity risk. Consequently, our model shows lower sequence preservation rates when compared back to the original murine antibody (Table 2).

This higher mutation count represents an intentional and beneficial trade-off. Our primary goal is to maximize human-likeness to reduce potential immunogenicity, not to minimally perturb the murine

Table 1: **Comparison of humanness performance.** The OASis value measures the absolute difference in OASis medium identity between humanized and parental antibodies. The T20 score reflects the absolute change between humanized and parental antibodies. Germline identity is the similarity between humanized antibodies and their most homologous human germline sequence, as measured by ANARCI (Dunbar & Deane, 2016). Mutation numbers represents the average sequence differences between humanized antibodies and their parental antibodies.

| Method | OASis (%) | T20 score (%) | | Germline identity (%) | | Mutation numbers | |
| --- | --- | --- | --- | --- | --- | --- | --- |
| | OASis | H | L | H | L | H | L |
| Experiment | 33.82 | 12.61 | 14.51 | - | - | 29.32 | 19.24 |
| Sapiens | 29.73 | 9.81 | 10.90 | 84.62 | 89.75 | 13.64 | 12.72 |
| Hu-mAb | 0.074 | 1.18 | 3.61 | 74.29 | 81.08 | 13.72 | 8.20 |
| Humatch | 18.02 | 8.45 | 5.98 | 83.16 | 87.02 | 16.56 | 14.36 |
| CUMAb | 37.73 | 12.85 | 11.85 | 93.96 | 94.78 | 24.05 | 20.90 |
| Hu-MCTs | **41.63** | **18.32** | **15.23** | **98.77** | **99.64** | 33.88 | 27.92 |

Table 2: **Sequence preservation and mutation precision.** The table reports sequence conservation (identity to parental antibody for total and Vernier residues) and mutation precision (agreement with experimental mutations).

Table 3: **VH-VL pairing quality.** Scores from the Humatch CNN-P model range from 0 (poor) to 1 (excellent).

| Method | Preservation (%) | | | | Mutation precision (%) | | | |
| --- | --- | --- | --- | --- | --- | --- | --- | --- |
| | Total | | Vernier | | Total | | Vernier | |
| | H | L | H | L | H | L | H | L |
| Experiment | 78.34 | 82.35 | 79.25 | 93.14 | - | - | - | - |
| Sapiens | 88.55 | 88.33 | 86.00 | 94.00 | 44.20 | 76.67 | 71.49 | 80.22 |
| Hu-mAb | 92.07 | 92.51 | 92.00 | 98.00 | 34.09 | 57.51 | 39.25 | 47.87 |
| Humatch | 88.85 | 86.86 | 92.43 | 91.14 | 42.16 | 50.00 | 62.54 | 57.87 |
| CUMAb | 79.92 | 80.90 | 82.81 | 92.14 | 35.44 | 38.89 | 53.37 | 56.14 |
| Hu-MCTs | 69.21 | 72.47 | 70.75 | 88.86 | 28.32 | 56.35 | 44.74 | 39.08 |

| Method | CNN-P Score |
| --- | --- |
| Experiment | 0.79 |
| Sapiens | 0.87 |
| Hu-mAb | 0.43 |
| Humatch | 0.95 |
| CUMAb | 0.58 |
| Hu-MCTs | **0.98** |

sequence. The fact that higher mutation counts correlate with top-tier humanness scores demonstrates that the modifications made by Hu-MCTs are highly targeted and effective. This approach prioritizes generating diverse, robustly humanized candidates that, as we show in Section 4.2.2, still maintain critical structural integrity and chain pairing quality.

#### 4.2.2 STRUCTURAL INTEGRITY AND CHAIN PAIRING QUALITY

Our model demonstrates a robust balance between high humanness and structural preservation. For structural stability, Hu-MCTs achieved comparable CDR1/CDR2 RMSD values to leading methods in both heavy and light chain regions (Table 4). For instance, its performance is on par with the structure-focused CUMAb in heavy chain CDR2 (0.627 Å all-atom RMSD) and Humatch in heavy chain CDR1 (0.226 Å backbone RMSD).

Notably, Hu-MCTs shows distinct fidelity patterns linked to our hierarchical MCTS reward function. In the heavy chain CDR3 (HCDR3)—the loop most critical for antigen binding—our method achieved the second-lowest all-atom RMSD (1.273 Å), surpassed only by the conservative Hu-mAb (1.105 Å). This is by design, as we assigned a higher weight to HCDR3 stability during optimization. In contrast, methods like CUMAb exhibited poorer HCDR3 preservation (1.508 Å), likely due to challenges in accurately modeling long CDR loops. While Hu-mAb had the lowest RMSD, this came at the cost of the lowest humanness score, highlighting a significant trade-off further discussed in Appendix C.2. Hu-MCTs successfully navigates this trade-off, achieving the highest humanness and Germline Identity despite introducing more mutations (Table 1).

Visual analysis (Figure 3) further reveals that while most methods maintain comparable HCDR3 structures, our model uniquely preserves the native conformation of the LCDR1 region, which showed significant deviations in competitor models. This indicates superior global structural fidelity. For VH-VL pairing quality, Hu-MCTs achieved a near-perfect score of 0.98, outperforming chain-independent methods like Hu-mAb (0.43) and CUMAb (0.58) (see Table 3). This compatibility stems from our model's joint optimization of both chains. To ensure the robustness of these structural findings, we performed a post-hoc validation using the high-fidelity AlphaFold3 predictor.

Table 4: **Structural RMSD comparison using Tfold.** Reported values are average all-atom and backbone RMSD (Å) for heavy/light chain CDRs, measured against murine parental antibodies. Lower values indicate better structural preservation.

| | | Heavy Chain | | | | Light Chain | |
|------|------------|-------------|---------------|------|------------|-------------|---------------|
| CDR | Method | All-atom (Å) | Backbone (Å) | CDR | Method | All-atom (Å) | Backbone (Å) |
| CDR1 | Experiment | 0.473 | 0.271 | CDR1 | Experiment | **0.627** | 0.271 |
| | Sapiens | 0.627 | 0.308 | | Sapiens | 0.782 | 0.334 |
| | Hu-mAb | 0.476 | **0.182** | | Hu-mAb | 0.695 | 0.297 |
| | Humatch | 0.490 | 0.226 | | Humatch | 0.711 | 0.294 |
| | CUMAb | 0.547 | 0.245 | | CUMAb | 0.744 | 0.286 |
| | Hu-MCTs | **0.442** | 0.252 | | Hu-MCTs | 0.663 | **0.271** |
| CDR2 | Experiment | 0.550 | 0.263 | CDR2 | Experiment | 0.279 | **0.065** |
| | Sapiens | 0.649 | 0.292 | | Sapiens | 0.387 | 0.094 |
| | Hu-mAb | **0.400** | **0.180** | | Hu-mAb | **0.204** | 0.072 |
| | Humatch | 0.457 | 0.191 | | Humatch | 0.312 | 0.090 |
| | CUMAb | 0.510 | 0.292 | | CUMAb | 0.236 | 0.074 |
| | Hu-MCTs | 0.461 | 0.221 | | Hu-MCTs | 0.299 | 0.076 |
| CDR3 | Experiment | 1.565 | 0.848 | CDR3 | Experiment | 0.634 | 0.277 |
| | Sapiens | 1.481 | 0.865 | | Sapiens | 0.718 | 0.318 |
| | Hu-mAb | **1.105** | **0.608** | | Hu-mAb | **0.547** | **0.227** |
| | Humatch | 1.371 | 0.820 | | Humatch | 0.601 | 0.278 |
| | CUMAb | 1.508 | 0.888 | | CUMAb | 0.641 | 0.269 |
| | Hu-MCTs | 1.273 | 0.776 | | Hu-MCTs | 0.623 | 0.265 |

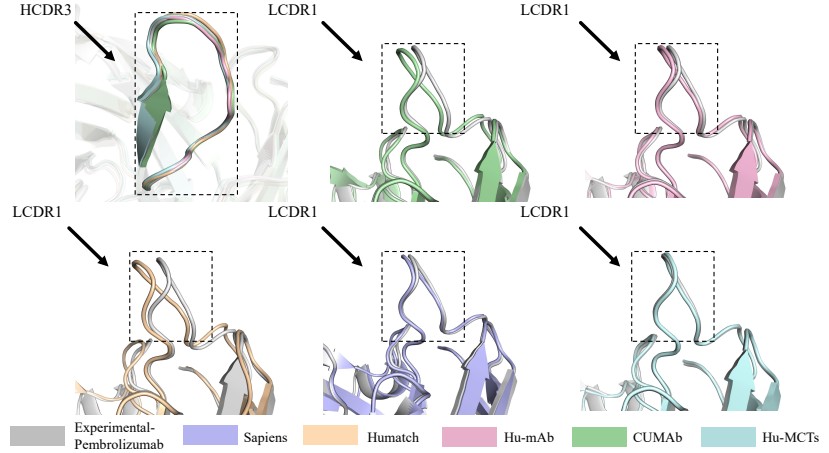

Figure 3: **CDR Structural Comparison in Humanized Antibodies.** Left (HCDR3): Structural alignment across methods. Right (LCDR1): Hu-MCTs (cyan) nearly overlaps with Experimental-Pembrolizumab (gray), while competitors (dashed boxes) show deviations. Our pipeline preserves LCDR1 structural fidelity better than others, while maintaining consistent HCDR3 alignment.

The results, detailed in the Appendix (Table 11), were highly consistent with our primary analysis and confirm the effectiveness of our method.

### 4.2.3 SEQUENCE PLAUSIBILITY ANALYSIS

Beyond geometric stability, we evaluated the **biological plausibility** of the generated sequences using AntiBERTy (Ruffolo et al., 2021), a language model that assesses how natural an antibody sequence is. Scores closer to zero indicate a higher likelihood of the sequence folding into a stable and functional antibody. As shown in Table 5, Hu-MCTs generated the most plausible sequences (-0.431), outperforming all baselines and even the experimentally validated antibodies. This result suggests that our method excels at capturing the subtle sequence patterns characteristic of functional human antibodies.

### 4.2.4 ABLATION STUDY OF STRUCTURAL STABILITY SCORING

To validate the role of structural stability in our pipeline, we conducted an ablation study by removing the $S_{\text{stab}}$ term from the MCTS reward function (see Table 6 and Table 7).

Table 5: **Average plausibility scores computed by AntiBERTy.** Higher scores (closer to zero) are better.

| Method | Average Plausibility Score |
|---|---|
| **Hu-MCTs (ours)** | **-0.431** |
| CUMAb | -0.516 |
| Experiment | -0.623 |
| Biophi | -0.639 |
| Humatch | -0.770 |
| Hu-mAb | -0.870 |

Table 6: **The impact of applying structural constraints on the humanness score.**

| Condition | OASis Score (%) | T20 Score (%) | | Germline Identity (%) | |
|---|---|---|---|---|---|
| | | H | L | H | L |
| With $S_{stab}$ | 41.63 | 18.32 | 15.63 | 98.77 | 99.64 |
| Without $S_{stab}$ | 43.37 | 20.04 | 16.79 | 99.21 | 99.23 |

Table 7: **Comparison of RMSD changes with and without the inclusion of structural scoring.**

| Condition | Chain Type | CDR1 (Å) | CDR2 (Å) | CDR3 (Å) |
|---|---|---|---|---|
| | H (All-atom) | 0.44 | 0.46 | 1.27 |
| With $S_{stab}$ | H (Backbone) | 0.22 | 0.25 | 0.78 |
| | L (All-atom) | 0.66 | 0.30 | 0.62 |
| | L (Backbone) | 0.27 | 0.08 | 0.27 |
| | H (All-atom) | 0.67 | 0.66 | 1.76 |
| Without $S_{stab}$ | H (Backbone) | 0.32 | 0.33 | 1.01 |
| | L (All-atom) | 0.92 | 0.44 | 0.81 |
| | L (Backbone) | 0.46 | 0.11 | 0.38 |

**Comparison of Humanness Metrics.** When $S_{stab}$ was removed, the OASis score increased from 41.63% to 43.37%, and both heavy and light of the T20 score showed moderate increases. However, the Germline Identity metrics exhibited divergent changes: heavy chain rose from 98.77% to 99.21%, while light chain slightly decreased from 99.64% to 99.23%. This suggests that removing structural constraints has a minor negative impact on light chain germline matching, but overall, the improvement in humanness metrics implies stronger tendency in sequence feature fitting.

**Structural Stability.** The results confirmed that excluding $S_{stab}$ led to notable increases in CDR RMSD values across all loops (e.g., heavy chain CDR1 all-atom RMSD rose from 0.44 Å to 0.67 Å, CDR3 backbone RMSD increased from 0.78 Å to 1.01 Å). The RMSD increases in CDR regions-especially CDR3, the core antigen-binding region-highlight the critical role of $S_{stab}$ in maintaining the structural integrity of antigen-binding sites. These results validate the necessity of structural scoring in the fidelity of their 3D structures: relying solely on humanness metrics may compromise structural stability, while incorporating $S_{stab}$ effectively avoids this trade-off.

## 5 CONCLUSION

In this work, we introduced **Hu-MCTs**, a novel framework for antibody humanization that systematically balances sequence humanness and structural integrity. By combining a generative model for paired heavy and light chains with a Monte Carlo Tree Search, our method performs multi-objective optimization directly in a learned latent space. This approach makes two key contributions: it is the first to apply this paradigm to the specific constraints of antibody humanization, and its joint optimization of both chains preserves critical VH-VL interactions often ignored by prior methods. Furthermore, by integrating structural stability directly into the optimization objective, our framework addresses a major gap in purely sequence-based approaches, yielding candidates with high biological plausibility. While our current work focuses on murine antibodies, the modularity of Hu-MCTs makes it adaptable for future extensions, including applicability to non-murine species and the integration of more advanced functional predictors like antigen-binding affinity.

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

# A    BACKGROUND

## A.1    PAIRED ANTIBODY DESIGN

The paired antibody optimization strategy in Hu-MCTs offers several key advantages. By jointly considering heavy and light chains, it captures the intricate VH-VL interactions that are crucial for antibody function (Chinery et al., 2024; Ma et al., 2024; Zhong, 2021). This holistic approach ensures that humanized antibodies exhibit both high humanness scores and preserved structural-functional integrity. For example, the analysis of VH-VL pairing quality, demonstrated that Hu-MCTs humanized antibodies closely resembled murine counterparts. This is in contrast to methods that optimize chains independently, which may disrupt these critical interactions and lead to reduced antigen binding affinity or stability.

## A.2    LATENT SPACE REPRESENTATIONS FOR EFFECTIVE SEARCH

In Hu-MCTs, the adoption of latent space for search is primarily driven by the exorbitant costs and inefficiencies associated with optimizing black-box functions within high-dimensional and highly-modal input spaces (Yu et al., 2024). By encoding antibody sequences into a latent space, we effectively reduce the dimensionality of the search space while capturing the underlying structural and sequence-relationships inherent to human antibodies (Reidenbach, 2024). This approach leverages the prior knowledge derived from the pre-training of the Conditional Variational Autoencoder (CVAE) on a diverse human antibody repertoire, steering the optimization process toward human-compatible sequences.

Moreover, similar to the concept in Bayesian Optimization (BO), where high-dimensional optimization is often transformed into low-dimensional exploration, the latent space provides a compressed representation. This enables efficient exploration of high-value input design patterns, circumventing the complex computations and sampling complexity issues prevalent in high-dimensional spaces. It addresses the challenges of inaccurate optimization trajectories or insufficient coverage in high-dimensional spaces (Xing et al., 2025), which are typical in traditional black-box optimization methods. Additionally, the ability to sample from the latent space facilitates the generation of a diverse array of humanized antibodies. This diversity significantly increases the likelihood of identifying optimal solutions, as it explores various modes within the latent space that correspond to different high-value input designs.

## A.3    BLACK-BOX OPTIMIZATION WITH MONTE CARLO TREE SEARCH

Antibody humanization requires the optimization of complex objectives such as humanness and structural stability, which arise from intricate sequence–structure–function relationships that cannot be modeled explicitly or differentiated. These objectives are thus treated as "black-box" functions, meaning their values can only be estimated through computational predictions (e.g., structure modeling with Tfold) or experimental measurements, without direct access to gradients.

Black-box optimization (BBO) methods are well suited to such settings where the objective function is either unknown or non-differentiable, and only input-output evaluations are accessible. These methods explore the solution space by balancing **exploration** (sampling diverse candidates to discover novel regions) and **exploitation** (focusing on promising variants to refine solutions). In our approach, we adopt **Monte Carlo Tree Search (MCTS)** as the core BBO algorithm. MCTS efficiently navigates the high-dimensional antibody sequence space using a multi-objective reward function that jointly considers humanness and structural stability, enabling the identification of optimized variants that balance both critical criteria.

The MCTS algorithm offers several advantages over traditional optimization methods. Firstly, it is a model-free optimization algorithm, which means that it does not require explicit knowledge of the objective function or the search space. This makes it highly flexible and applicable to a wide range of problems. Secondly, MCTS is able to balance exploration and exploitation effectively. By using the Upper Confidence Bound (UCB) score to select nodes for expansion, it is able to explore promising regions of the search space while also exploiting known good solutions. This helps to avoid getting trapped in local optima and increases the chances of finding globally optimal solutions. Finally,

MCTS is a computationally efficient algorithm, which makes it suitable for large-scale optimization problems.

## B    METHODOLOGICAL DETAILS AND JUSTIFICATIONS

### B.1    THEORETICAL ANALYSIS OF SAMPLE EFFICIENCY

The synergy between our CVAE-based latent space projection and MCTS-guided search provides a significant advantage in sample efficiency. The challenge of humanization can be framed as a black-box optimization problem where the expected simple regret $R(T)$ after $T$ evaluations in a $d$-dimensional space is bounded by (Yang et al., 2022):

$$\mathbb{E}[R(T)] = \mathcal{O}\left(C_{\text{good}} \sqrt[d]{\ln T} \; T^{\frac{d-1}{d}}\right) \tag{3}$$

where $C_{\text{good}}$ is a constant reflecting the concentration of high-quality solutions. Our approach enhances search efficiency in three key ways:

**1. Latent-space projection reduces sample complexity.** By projecting the high-dimensional antibody sequence space ($d \approx 180$) into a low-dimensional CVAE latent space ($d' \approx 30$), we immediately tighten the regret bound in Equation 3. This substitution of $d$ with $d'$ shrinks the exponent from $\frac{d-1}{d} \approx 0.994$ to $\frac{d'-1}{d'} \approx 0.967$, leading to an exponential reduction in the sample complexity required to find an $\varepsilon$-accurate solution from $\mathcal{O}(\varepsilon^{-180})$ to a more tractable $\mathcal{O}(\varepsilon^{-30})$.

**2. The CVAE endows the latent geometry with biochemical priors.** The CVAE organizes the latent space to reflect crucial biochemical properties. First, through **neighborhood preservation**, sequences decoding to structurally similar antibodies are mapped to nearby points, creating a smoother optimization landscape. Second, by training on a vast human antibody repertoire, the CVAE enforces **implicit constraints**, biasing the search toward human-compatible designs. These learned priors effectively reduce the $C_{\text{good}}$ constant, further accelerating convergence.

**3. MCTS performs adaptive space-partitioning.** MCTS acts as an efficient adaptive algorithm on this structured latent space, intelligently balancing exploration and exploitation. This is critical for navigating the highly multimodal fitness landscape of antibody design, allowing the search to escape local optima and converge to high-quality variants with far fewer evaluations than required in the original discrete sequence space.

### B.2    JUSTIFICATION FOR STRUCTURE PREDICTOR SELECTION

The choice of a structure prediction tool within an iterative framework like MCTS requires a trade-off between accuracy and speed. While state-of-the-art models like AlphaFold3 offer high accuracy, their inference time is prohibitive for the hundreds of evaluations in a single MCTS run. We selected **Tfold** for its favorable balance, as it achieves competitive accuracy for antibody structures—particularly CDR loops—with significantly faster inference times. This choice was critical for computational feasibility. To validate this approach, we performed post-hoc evaluations with AlphaFold3, confirming that the structural trends identified using Tfold were consistent with those from a higher-accuracy predictor (see Section D.2).

### B.3    RATIONALE FOR FOCUSING ON MURINE ANTIBODIES

Our work concentrates on murine-to-human humanization, a decision guided by three practical considerations:

- **Consistency with Prior Research:** The majority of foundational humanization studies have used murine candidates, establishing a rich set of benchmarks essential for contextualizing our work.

- **Clinical Relevance and Data Availability:** Murine antibodies remain a primary source of preclinical therapeutics, leading to large-scale public datasets (e.g., OAS) that are invaluable for model development and validation.

- **Limitations of Non-Murine Data:** Antibodies from other species (e.g., camelids) have limited public data and often exhibit distinct CDR architectures, posing significant out-of-distribution risks for models trained on human data.

### B.4 APPLICABILITY TO NON-MURINE ANTIBODIES

Our framework is trained exclusively on human antibody sequences, making it species-agnostic in principle. However, extending it to non-murine antibodies presents challenges. The limited availability of data and distinct CDR architectures (e.g., longer CDR3s in camelid antibodies) create distributional differences that could impair performance. Adapting our framework to other species is a key direction for future work and will likely require fine-tuning with species-specific data or applying more advanced transfer learning techniques.

## C ANALYSIS OF HUMANIZATION STRATEGY

### C.1 HUMANIZATION STRATEGY AND IMMUNOGENICITY RISK

A key finding is that Hu-MCTs introduces more mutations than conservative baselines. This is a deliberate strategic choice. Methods that prioritize minimal changes (e.g., simple CDR grafting) risk leaving residual murine epitopes intact, which can trigger an immune response. In contrast, our approach performs a dynamic search in a learned latent space to identify solutions that optimally balance humanness, CDR compatibility, and structural stability. The resulting mutations are targeted modifications that refine the initial human frameworks to better accommodate the murine CDRs while maximizing alignment with the broader landscape of human antibodies. This comprehensive optimization reduces immunogenicity risk more effectively than simply minimizing mutation counts, a claim supported by the high germline identity and sequence plausibility scores our method achieves.

### C.2 DETAILED COMPARISON WITH HU-MAB

The performance disparity between Hu-MCTs and Hu-mAb highlights their different humanization philosophies. Hu-mAb prioritizes **maximum structural preservation through minimal mutations**, leading to low RMSD but also **limited humanness**, as it cannot fully eliminate murine sequence features (Table 8).

In contrast, Hu-MCTs is engineered to prioritize **humanness enhancement without sacrificing structural stability**. It introduces more mutations, but these changes are guided by a multi-objective function to co-optimize for human-likeness and structural integrity. This strategy allows Hu-MCTs to effectively remove murine epitopes and achieve state-of-the-art humanness scores while confining structural deviations to non-essential regions. This balanced approach is critical for therapeutic antibodies, where minimizing immunogenicity and retaining efficacy are paramount.

Table 8: Quantitative comparison of Hu-MCTs and Hu-mAb, highlighting the trade-off between humanness and mutation count.

| Method | OASis (%) | T20 score (%) | | Germline identity (%) | | Mutation numbers | |
|---|---|---|---|---|---|---|---|
| | | H | L | H | L | H | L |
| Hu-mAb | 0.074 | 1.18 | 3.61 | 74.29 | 81.08 | 13.72 | 8.20 |
| **Hu-MCTs** | **41.63** | **18.32** | **15.23** | **98.77** | **99.64** | 33.88 | 27.92 |

## D ADDITIONAL EXPERIMENTAL ANALYSIS

### D.1 HYPERPARAMETER ABLATION STUDY: IMPACT OF K

To assess the robustness of our method, we analyzed the impact of the number of initial starting points, $K$, with $K \in \{10, 20, 30, 50\}$. The results show that core humanness metrics remain stable

Table 9: Impact of the number of starting points ($K$) on humanness metrics.

| K value | OASis (%) | T20 Score H (%) | T20 Score L (%) | Germline Identity H (%) | Germline Identity L (%) | Mutation Num H | Mutation Num L |
|---|---|---|---|---|---|---|---|
| 10 | 40.52 | 17.75 | 14.82 | 96.58 | 97.32 | 33.12 | 27.28 |
| 20 | 41.05 | 18.02 | **15.25** | 97.86 | 98.95 | 33.56 | 27.69 |
| 30 | 41.32 | 18.15 | 15.08 | 98.21 | 99.12 | **33.75** | 27.81 |
| 50 | **41.63** | **18.32** | 15.23 | **98.77** | **99.64** | 33.82 | **27.92** |

Table 10: Impact of K on structural preservation (RMSD, Å). Best results for each metric are in **bold**.

| | | Heavy Chain | | | | Light Chain | |
|---|---|---|---|---|---|---|---|
| CDR | K Value | All-Atom RMSD (Å) | Backbone RMSD (Å) | CDR | K Value | All-Atom RMSD (Å) | Backbone RMSD (Å) |
| CDR1 | K = 10 | 0.453 | 0.258 | CDR1 | K = 10 | 0.678 | 0.276 |
| | K = 20 | 0.445 | 0.254 | | K = 20 | 0.666 | **0.270** |
| | K = 30 | 0.444 | 0.253 | | K = 30 | 0.665 | 0.271 |
| | K = 50 | **0.442** | **0.252** | | K = 50 | **0.663** | 0.271 |
| CDR2 | K = 10 | 0.472 | 0.226 | CDR2 | K = 10 | 0.306 | 0.079 |
| | K = 20 | 0.464 | 0.223 | | K = 20 | 0.302 | 0.077 |
| | K = 30 | **0.458** | 0.222 | | K = 30 | 0.300 | **0.074** |
| | K = 50 | 0.461 | **0.221** | | K = 50 | **0.299** | 0.076 |
| CDR3 | K = 10 | 1.305 | 0.798 | CDR3 | K = 10 | 0.638 | 0.271 |
| | K = 20 | 1.286 | 0.785 | | K = 20 | 0.630 | 0.268 |
| | K = 30 | 1.278 | 0.779 | | K = 30 | 0.626 | 0.266 |
| | K = 50 | **1.273** | **0.776** | | K = 50 | **0.623** | **0.265** |

across these values (Table 9), demonstrating that the method is not overly sensitive to this hyperparameter. Similarly, while increasing $K$ leads to marginal improvements in structural preservation (Table 10), the benefit quickly diminishes. We chose $K = 50$ for our main experiments as a robust setting that balances exploration and computational cost, and we recommend $K \geq 10$ for general use.

### D.2 INDEPENDENT STRUCTURAL VALIDATION WITH ALPHAFOLD3

To independently verify our structural preservation results, we performed a post-hoc analysis on 60 randomly selected variants using the high-accuracy AlphaFold3 model. The results, presented in Table 11, are highly consistent with the Tfold-based findings in the main text. Hu-MCTs consistently achieves low RMSD values across all CDRs and performs favorably against all baselines, confirming the effectiveness of our structure-aware optimization strategy.

### D.3 COMPUTATIONAL RESOURCES AND EFFICIENCY

Experiments were conducted on a single NVIDIA RTX 4090 GPU (48GB). Humanizing one paired antibody takes approximately **50 minutes** ($\sim$3000s) with our default settings. Direct runtime comparisons with baselines are challenging, as most are web servers with unknown hardware. However, as shown in Table 12, our method's runtime is practical for research settings. While more intensive than sequence-only methods, the cost is justified by the more reliable, structure-aware outcomes.

## E TRAINING DETAILS

### E.1 TRAINING DATASET AND PREPROCESSING

Our training dataset was sourced from the Observed Antibody Space (OAS), from which we collected approximately 1.74 million paired human antibody sequences. To create a uniform input for our models, we standardized all sequences using the IMGT numbering scheme, resulting in heavy chains of 152 residues and light chains of 139 residues. This dataset was then partitioned into a 99% training set and a 1% validation set to ensure robust model training while monitoring for overfitting.

Table 11: Structural RMSD validation using AlphaFold3. Reported values are the average all-atom and backbone RMSD (Å) for each CDR, measured against the parental murine antibodies. Best results are in **bold**, and second best are underlined.

| | | Heavy Chain | | | | Light Chain | | |
|------|----------|-------------|--------------|------|----------|-------------|--------------|
| CDR | Method | All-atom (Å) | Backbone (Å) | CDR | Method | All-atom (Å) | Backbone (Å) |
| CDR1 | Experiment | 0.590 | 0.189 | CDR1 | Experiment | 0.688 | 0.186 |
| | Sapiens | 0.646 | 0.263 | | Sapiens | 0.650 | 0.221 |
| | Hu-mAb | **0.494** | **0.137** | | Hu-mAb | **0.469** | 0.176 |
| | Humatch | 0.630 | 0.238 | | Humatch | 0.702 | 0.404 |
| | CUMAb | 0.694 | 0.204 | | CUMAb | 0.603 | **0.157** |
| | Hu-MCTs | 0.541 | 0.209 | | Hu-MCTs | 0.531 | 0.162 |
| CDR2 | Experiment | 0.443 | 0.125 | CDR2 | Experiment | 0.135 | 0.029 |
| | Sapiens | 0.407 | 0.150 | | Sapiens | 0.176 | 0.028 |
| | Hu-mAb | 0.365 | **0.081** | | Hu-mAb | **0.122** | 0.027 |
| | Humatch | **0.312** | 0.092 | | Humatch | 0.191 | 0.028 |
| | CUMAb | 0.513 | 0.184 | | CUMAb | 0.133 | **0.021** |
| | Hu-MCTs | 0.385 | 0.113 | | Hu-MCTs | 0.149 | 0.026 |
| CDR3 | Experiment | 1.677 | 0.790 | CDR3 | Experiment | 0.635 | 0.188 |
| | Sapiens | 1.594 | 0.685 | | Sapiens | 0.745 | 0.187 |
| | Hu-mAb | **1.221** | 0.629 | | Hu-mAb | **0.459** | **0.107** |
| | Humatch | 1.495 | 0.643 | | Humatch | 0.509 | 0.173 |
| | CUMAb | 1.315 | 0.727 | | CUMAb | 0.479 | 0.177 |
| | Hu-MCTs | 1.304 | **0.528** | | Hu-MCTs | 0.643 | 0.136 |

Table 12: Comparison of humanization runtime for a paired heavy and light chain.

| Method | Approx. Humanization Time (s) |
|--------|-------------------------------|
| CUMAb | ∼24,688 (web) |
| Hu-mAb | ∼1,093 (web) |
| Hu-match | ∼33 (local) |
| Sapiens | ∼3 (web) |
| **Hu-MCTs (ours)** | **∼3,072 (local)** |

## E.2 PARAMETER-EFFICIENT TRAINING WITH LORA

Both the PLM and CVAE were trained using Low-Rank Adaptation (LoRA) (Hu et al., 2021), a parameter-efficient fine-tuning technique. Instead of training all model parameters, LoRA freezes the pre-trained weights and injects small, trainable low-rank matrices into the attention modules. This strategy significantly reduces the number of trainable parameters and computational overhead, while effectively adapting the models to the humanization task and mitigating the risk of overfitting.

## E.3 TRAINING CONFIGURATIONS

The specific configurations for our models, training hyperparameters, and LoRA are detailed below.

### E.3.1 MODEL ARCHITECTURES

The architectural details for the Protein Language Model (IgBERT) and the CVAE are summarized in Table 13.

### E.3.2 TRAINING HYPERPARAMETERS

The models were trained on NVIDIA H100 GPUs with the hyperparameters listed in Table 14.

### E.3.3 LORA CONFIGURATION

For both models, we applied a LoRA configuration with a rank ($r$) of 8, a scaling factor ($\alpha$) of 32, and a dropout rate of 0.05. No bias terms were trained.

Table 13: **Key Configuration Parameters of the PLM and CVAE Models.**

| Parameter | PLM (IgBERT) | CVAE (T5-based) |
|---|---|---|
| *Network Architecture* | | |
| Hidden Size ($d_{\text{model}}$) | 1024 | 1024 |
| Intermediate Size ($d_{\text{ff}}$) | 4096 | 16384 |
| Number of Layers | 30 (hidden) | 24 (encoder), 24 (decoder) |
| Number of Attention Heads | 16 | 32 |
| *Vocabulary & Sequences* | | |
| Vocabulary Size | 30 | 144 |
| Max Position Embeddings | 40000 | 512 |
| Position Encoding Type | Absolute | Relative |
| *Regularization & Numerics* | | |
| Activation Function | `gelu` | `relu` |
| Dropout Rate | 0.0 | 0.1 |
| Layer Norm Epsilon | $1 \times 10^{-12}$ | $1 \times 10^{-6}$ |

Table 14: **Training Hyperparameters and Hardware Configuration.**

| Parameter | PLM (IgBERT) | CVAE (T5-based) |
|---|---|---|
| *Training Settings* | | |
| Batch Size | 256 | 32 |
| Training Epochs | 30 | 3 |
| Learning Rate | $1 \times 10^{-5}$ | $2 \times 10^{-5}$ |
| *Hardware and Duration* | | |
| GPU Type | H100 | H100 |
| Number of GPUs | 4 | 2 |
| Training Duration | ~48 hours | ~140 hours |

## F  LIMITATIONS AND FUTURE DIRECTIONS

Despite its promising performance, Hu-MCTs has several limitations that provide clear avenues for future research. Our current framework is tailored for the humanization of murine variable regions (VH/VL), a focus guided by the rich availability of benchmark data and the clinical relevance of murine-derived antibodies. However, this scope does not account for constant regions and limits direct applicability to non-murine species. A primary goal for future work is therefore to broaden this applicability by adapting the model for species like camelids and rabbits—likely through transfer learning—and by incorporating constant regions to enable the end-to-end design of full-length antibodies.

Beyond expanding the model's scope, we also aim to enhance its predictive capabilities. The current model evaluates candidates based on static structures and does not explicitly predict antigen-binding affinity or conformational dynamics. This is a pragmatic choice reflecting the current state of the field, where reliable, high-throughput predictors for these complex properties are not yet available. Fortunately, the framework's modularity is a key advantage that will facilitate future improvements. As more sophisticated functional predictors mature, we plan to integrate emerging tools capable of assessing epitope-binding interactions and conformational dynamics, which will better preserve antibody function and efficacy post-humanization.

While these extensions promise a more powerful framework, we are mindful of computational demands. As a structure-aware method, Hu-MCTs is inherently more computationally intensive than sequence-only approaches—a necessary trade-off for achieving higher structural fidelity. By continuing to refine this balance, the core MCTS algorithm can be enhanced to manage an even wider array of competing therapeutic goals, such as expression yield and other developability properties. Evolving the optimization framework in this manner will transform Hu-MCTs into a more compre-

hensive and versatile tool, opening up its application to diverse antibody engineering tasks such as affinity maturation and the design of antibody-drug conjugates.

## G  BROADER IMPACT

The development of Hu-MCTs has several scientific implications. Firstly, it provides a new approach for antibody humanization that is more effective and efficient than existing methods. This could lead to the development of more safe and effective antibody-based therapeutics. Secondly, it provides a new tool for understanding the structural and functional properties of antibodies. By exploring the sequence space of human antibodies, Hu-MCTs can help to identify the key residues and motifs that are important for antibody function and immunogenicity.

The development of Hu-MCTs also has several clinical implications. Firstly, it could lead to the development of safer and effective antibody-based therapeutics. By reducing the immunogenicity of murine antibodies, Hu-MCTs can help to improve the safety and efficacy of antibody-based therapies. Finally, it could lead to the development of new antibody-based therapies for diseases that are currently difficult to treat. By exploring the sequence space of human antibodies, Hu-MCTs can help to identify new antibodies that have the potential to treat these diseases.

## H  ALGORITHM PSEUDOCODE

---

**Algorithm 2** Partition Function

---

**Require:** Input Space $\Omega$, Samples $\mathcal{S}_t$, Node partition threshold $N_{\text{thres}}$, Partitioning Latent Model $s(\mathbf{x})$
1: Set $\mathcal{V}_0 = \{\Omega\}$
2: Set $\mathcal{V}_{\text{queue}} = \{\Omega\}$
3: **while** $\mathcal{V}_{\text{queue}} \neq \emptyset$ **do**
4:     $\Omega_p \leftarrow \mathcal{V}_{\text{queue}}.\text{pop}(0)$
5:     **if** $n(\Omega_p) \geq N_{\text{thres}}$ **then**
6:         $S_{\text{good}}, S_{\text{bad}} \leftarrow$ samples from $\mathcal{S}_t$ corresponding to indices of $k\text{-means}(s(\Omega_p \cap \mathcal{S}_t))$
7:         Fit SVM on $S_{\text{good}}, S_{\text{bad}}$
8:         Use SVM to split $\Omega_p$ into $\Omega_{\text{good}}, \Omega_{\text{bad}}$
9:         $\mathcal{V}_0 \leftarrow \mathcal{V}_0 \cup \{\Omega_{\text{good}}, \Omega_{\text{bad}}\}$
10:        $\mathcal{V}_{\text{queue}} \leftarrow \mathcal{V}_{\text{queue}} \cup \{\Omega_{\text{good}}, \Omega_{\text{bad}}\}$
11:     **end if**
12: **end while return** $\mathcal{V}_0$

---

---

**Algorithm 3** CMA-ES: Covariance Matrix Adaptation Evolution Strategy

---

**Require:**
1: Objective function $f : \mathbb{R}^n \to \mathbb{R}$,
2: Population size $\lambda$,
3: Initial mean $\mathbf{m} \in \mathbb{R}^n$,
4: Initial covariance matrix $\mathbf{C} \in \mathbb{R}^{n \times n}$,
5: Initial step size $\sigma \in \mathbb{R}_+$,
6: Number of iterations $T$,
7: Mean update size $\mu = \lfloor \lambda/2 \rfloor$
**Ensure:** Optimized mean $\mathbf{m}$, All evaluated states $\mathcal{Z}$
8: Initialize evolution paths: $\mathbf{p}_\sigma \leftarrow \mathbf{0}$, $\mathbf{p}_c \leftarrow \mathbf{0}$
9: Initialize state memory: $\mathcal{Z} \leftarrow \emptyset$
10: **for** generation $t = 1$ **to** $T$ **do**
11: $\quad \mathcal{Y} \leftarrow \emptyset, \mathcal{V} \leftarrow \emptyset$
12: $\quad$ **for** $i = 1$ **to** $\lambda$ **do**
13: $\quad\quad$ Sample $\mathbf{z}_i \sim \mathcal{N}(\mathbf{0}, \mathbf{I})$
14: $\quad\quad$ Generate offspring: $\mathbf{y}_i \leftarrow \mathbf{m} + \sigma \mathbf{B} \mathbf{D} \mathbf{z}_i$
15: $\quad\quad$ Evaluate fitness: $v_i \leftarrow f(\mathbf{y}_i)$
16: $\quad\quad$ Store results: $\mathcal{Y} \leftarrow \mathcal{Y} \cup \{\mathbf{y}_i\}, \mathcal{V} \leftarrow \mathcal{V} \cup \{v_i\}$
17: $\quad$ **end for**
18: $\quad$ Sort population: $\mathcal{Z} \leftarrow \text{sort}\left((\mathcal{Y}, \mathcal{V}), \text{by } v_i\right)$
19: $\quad$ Select top $\mu$ offspring: $\mathcal{Y}_\mu \leftarrow \mathcal{Z}[1 : \mu]$
20: $\quad$ Update mean:

$$\mathbf{m}' \leftarrow \mathbf{m} + \sum_{i=1}^{\mu} w_i (\mathbf{y}_{i:\lambda} - \mathbf{m})$$

21: $\quad$ Update evolution paths:

$$\mathbf{p}_\sigma \leftarrow (1 - c_\sigma) \mathbf{p}_\sigma + \sqrt{c_\sigma (2 - c_\sigma) \mu_{\text{eff}}} \mathbf{B} \mathbf{D}^{-1} \frac{\mathbf{m}' - \mathbf{m}}{\sigma}$$

$$\mathbf{p}_c \leftarrow (1 - c_c) \mathbf{p}_c + \sqrt{c_c (2 - c_c) \mu_{\text{eff}}} \frac{\mathbf{m}' - \mathbf{m}}{\sigma}$$

22: $\quad$ Update covariance matrix:

$$\mathbf{C} \leftarrow (1 - c_1 - c_\mu) \mathbf{C} + c_1 \mathbf{p}_c \mathbf{p}_c^\top + c_\mu \sum_{i=1}^{\mu} w_i \mathbf{y}_i \mathbf{y}_i^\top$$

23: $\quad$ Adapt step size:

$$\sigma \leftarrow \sigma \cdot \exp\left( \frac{c_\sigma}{d_\sigma} \left( \frac{\|\mathbf{p}_\sigma\|}{E[\|\mathcal{N}(0, I)\|]} - 1 \right) \right)$$

24: $\quad$ Update mean: $\mathbf{m} \leftarrow \mathbf{m}'$
25: **end for**
26: **return** $\mathbf{m}, \mathcal{Z}$

---

