# OpenReview forum: "Structure-Aware Antibody Humanization via a Novel Black-box Optimization Algorithm"
_ICLR.cc/2026/Conference — Submitted to ICLR 2026_

### Official Review · Reviewer_b77H · 2025-10-30

**Soundness:** 2
**Presentation:** 1
**Contribution:** 2
**Rating:** 2
**Confidence:** 4

**Summary:**

This paper proposes a pipeline that combines PLM-guided initialization, CVAE-based generation, MCTS global search, and CMA-ES local optimization for antibody humanization. Experimental results demonstrate that the proposed method achieves higher humanness scores while better preserving CDR structural stability.

**Strengths:**

1.	It clearly clarifies the key issues in current antibody humanization work and proposes an integrated pipeline to improve antibody humanness while preserving structural integrity.
2.	The problem of antibody humanization is crucial for real-world therapeutic applications.

**Weaknesses:**

1.	Although I appreciate the real-world value of this work, the paper reads primarily as an application study that integrates existing AI techniques for use in the antibody domain. As such, it may be more suitable for a journal focused on antibodies or biotherapeutics, also given that all of the compared baselines are drawn from the bioscience literature.
2.	The writing of the technical part is not clear. Several symbols used in the algorithms are undefined, making the methods difficult to fully understand. In addition, the section describing the use of MCTS for latent space search (Lines 206–215) is hard to follow.
3.	The proposed method considers the structural integrity mainly by using the reward of structural stability. I am concerned that the paper may be overlooking more advanced sequence–structure co-design frameworks.

**Questions:**

1.	For training CVAE, are the $x$ and $c$ both extracted from the human sequences? If so, would it be a generalization issue when applied such a trained CVAE to humanization where $c$ would be from the murine antibody’s CDR.

---

> ### Author Response · Authors · 2025-12-02
>
> We thank the reviewer for their critical assessment and for acknowledging the crucial real-world value of antibody humanization. We appreciate the feedback on presentation and methodology,we will rewrite the section describing the MCTS latent space search to improve readability, ensuring the it is clearly explained. While we understand the concern regarding the "application" nature of the work, we respectfully argue that adapting latent space optimization (MCTS + CVAE) to solve a constrained, biological problem represents a meaningful contribution to the "AI for Science" domain, which is a interest of ICLR. We address the specific concerns below.
>
> **Q1: CVAE Generalization (Training on Human vs. Inference with Murine CDRs)**
>
> **Response:**
> The reviewer raises a valid concern regarding potential distribution shifts. However, the generalization gap is much smaller than it might appear due to the evolutionary relationship between murine and human antibodies.
> * **Structural Homology:** Unlike other distinct species, murine and human antibodies share significant **structural homology** in their variable regions. They evolved from common ancestors, and their CDR loops generally fall into a limited set of conserved 3D geometries known as **"canonical structures"** .
> * **Manifold Compatibility:** Although the *sequences* differ, the structural "grammar" of murine CDRs is highly compatible with the human framework rules learned by the CVAE. The CVAE is trained to learn the *compatibility* between FRs and valid CDR geometries. Since murine CDRs largely adhere to these valid geometries, the trained encoder can effectively project them into the latent space without suffering from the "out-of-distribution" failure modes that might occur with structurally distinct species (e.g., camelid VHH).

---

### Official Review · Reviewer_jCmW · 2025-11-01

**Soundness:** 3
**Presentation:** 3
**Contribution:** 3
**Rating:** 8
**Confidence:** 4

**Summary:**

This paper introduces Hu-mAb, a novel model for antibody humanization that preserves structural stability. Hu-mAb integrates an antibody protein language model with a black-box optimization strategy based on Monte Carlo Tree Search (MCTS). The model is first pretrained on extensive datasets of human antibody sequences to capture human-like sequence representations. During optimization, MCTS employs a composite reward function that jointly balances sequence human-likeness and structural stability. Comprehensive evaluations across multiple datasets demonstrate that Hu-mAb effectively humanizes antibodies while maintaining their structural integrity.

**Strengths:**

This paper effectively integrates protein language modeling with optimization techniques, addressing the issue of structural stability degradation observed in previous approaches. The proposed model jointly optimizes for both human-likeness and structural stability, enabling balanced and biologically meaningful antibody design. Moreover, this multi-objective optimization framework holds promise for extension to additional antibody properties, paving the way for more versatile and comprehensive antibody engineering.

**Weaknesses:**

The paper shows the effectiveness of antibody humanization. However, most of the evaluations are humanization for murine antibodies. The humanization ability of other types of antibodies remains unknown.

**Questions:**

1. In Table 2, it seems many baseline methods have higher mutation precision than Hu-MCT. Could the author provide more explanation on this?
2. Table 12 shows that the humanization runtime of Hu-MCTs is significantly slower than that of many baselines. Could the author discuss this tradeoff between runtime and accuracy in more detail?

---

> ### Author Response · Authors · 2025-12-02
>
> We are deeply grateful for the reviewer’s positive assessment and for recognizing the novelty of our framework and the promise of the multi-objective optimization design. We appreciate insightful questions regarding mutation precision and runtime tradeoffs. We address these points below.
>
> **Q1: Explanation for Lower Mutation Precision Relative to Mutation Count**
>
> The lower "mutation precision" (agreement with experimental mutations) in Hu-MCTs is not a failure to learn, but rather a deliberate trade-off resulting from our aggressive strategy to maximize humanness and reduce immunogenicity. This can be explained by three key factors:
>
> * **Optimization vs. Imitation:** The "mutation precision" metric rewards methods that strictly mimic historical experimental data. However, our goal is **not to copy** existing experiments, but to **outperform** them. Baselines often use conservative strategies (e.g., rigid CDR grafting) that miss optimization opportunities. In contrast, Hu-MCTs explores a broader latent sequence space to find novel solutions with significantly higher humanness scores (OASis/T20) that simply differ from the "ground truth" of older experiments.
> * **Impact of Mutation Volume:** There is a distinct difference in the quantity of modifications. For example, the baseline Hu-mAb introduces only ~13 mutations on the heavy chain, whereas Hu-MCTs introduces ~33. Because we modify a wider array of residues to thoroughly remove murine epitopes, our sequences statistically diverge more from conservative experimental targets, naturally lowering the precision score while improving the biological safety profile.
>
> **Q2: Tradeoff between runtime and accuracy**
>
> We acknowledge that Hu-MCTs is computationally more intensive than sequence-only baselines. This is the necessary cost for achieving **structure-aware** optimization, which ensures higher structural integrity than faster, sequence-based methods.
>
> * **Current Bottlenecks:** Our current inference time is distributed approximately as follows:
>     * **Sequence Evaluation (including Tfold structural prediction):** ~30%
>     * **CVAE Encoding/Decoding:** ~25%
>     * **MCTS Search:** ~35%
>     * **Overhead/Other:** ~10%
>
> * **Optimization Potential:** It is important to note that the current runtime reflects a research-grade implementation where the code efficiency for CVAE and MCTS has not yet been fully optimized.
> * **Future Plan:** We are actively refactoring the code to improve the efficiency of the latent space search and encoding steps. We believe we can significantly reduce the runtime in the final release without compromising the accuracy or structural stability benefits of the model.

---

### Official Review · Reviewer_B9kK · 2025-11-09

**Soundness:** 3
**Presentation:** 3
**Contribution:** 3
**Rating:** 6
**Confidence:** 3

**Summary:**

This paper introduces **Hu-MCTs**, a structure-aware antibody humanization framework that combines:
- **Protein Language Model (PLM)** for initialization,
- **Conditional Variational Autoencoder (CVAE)** for paired heavy/light chain latent representation,
- **Monte Carlo Tree Search (MCTS)** with CMA-ES for multi-objective optimization.

The key contribution lies in optimizing humanness and structural integrity simultaneously in latent space, explicitly penalizing CDR conformational disruption. Experiments on Humab25 and HuAb348 datasets show improvements over baselines (Sapiens, Hu-mAb, Humatch, CUMAb) in humanness scores (OASis, T20, germline identity) and structural RMSD.

**Strengths:**

- **Novelty**:
The integration of Monte Carlo Tree Search (MCTS) with CVAE-based latent space sampling for antibody humanization is novel and well-motivated. The multi-objective design that balances humanness and structural stability is a strong contribution.

- **Comprehensive benchmarking**:
The study compares Hu-MCTs against five baselines (Sapiens, Hu-mAb, Humatch, CUMAb, and experimental sequences) across multiple metrics including humanness, RMSD, pairing quality, and plausibility. Thorough comparision with the baselines clarifies the effectiveness of the proposed framework.

**Weaknesses:**

1. **Lack of structural evaluation**:
The study does not evaluate whether the humanized antibodies maintain antigen-binding functionality. While structural RMSD is used as a proxy for stability, it does not guarantee binding affinity. Including computational affinity prediction would strengthen the practical relevance of the results.

2. **Lack of inference time evalutation**:
The framework does not report inference time breakdowns for each stage of the pipeline (e.g., PLM initialization, CVAE encoding, MCTS search, CMA-ES refinement, and structure prediction). This omission makes it difficult to assess computational bottlenecks and scalability in practical settings.

**Questions:**

1. **Structural evaluation**: Could the authors provide an analysis of antigen-binding affinity through computational prediction tools such as AlphaBind [1]? How well do the RMSD of CDRs correlate with binding affinity?

2. **Inference time evaluation**: Could the authors provide a breakdown of inference time across different stages of the pipeline? and how does each stage scales with antibody sequence length?

3. **Reward function sensitivity**: How sensitive is the MCTS reward function to the weighting parameters? It would be helpful to include a trade-off graph between humanness and structural metrics by varying the reward weighting parameters. This would help validate the robustness of the multi-objective design.

___
[1] Agarwal, A. A., Harrang, J., Noble, D., McGowan, K. L., Lange, A. W., Engelhart, E., ... & Emerson, R. O. (2025, December). AlphaBind, a domain-specific model to predict and optimize antibody–antigen binding affinity. In Mabs (Vol. 17, No. 1, p. 2534626). Taylor & Francis.

---

> ### Author Response · Authors · 2025-12-02
>
> We thank the reviewer for their constructive feedback and for recognizing the novelty of integrating MCTS with CVAE-based latent space sampling, as well as the comprehensiveness of our benchmarking. We address the specific questions below to clarify our current constraints and future improvement plans.
>
> **Q1: Structural evaluation**
>
> We fully agree with the reviewer that maintaining antigen-binding affinity is the ultimate goal of humanization and that RMSD is a structural proxy.
> * **Data Constraints:** Our current work relies primarily on the OAS and Humab datasets, which focus on antibody variable regions but unfortunately lack paired antigen  data for the vast majority of sequences. This limits our ability to directly apply antigen-dependent affinity predictors (like AlphaBind) at this stage.
> * **Future Plan:** We are actively working on curating a dataset subset that includes paired antigen-antibody structures. We plan to incorporate explicit affinity prediction experiments in the next version of the framework to bridge the gap between structural stability (RMSD) and functional binding.
>
> **Q2: Inference time evaluation**
>
> We appreciate the suggestion to analyze computational bottlenecks.
> * **Time Breakdown:** Based on our current implementation, the approximate breakdown of inference time is as follows:
>     * **Sequence Evaluation (including Tfold structural prediction):** ~30%
>     * **CVAE Encoding/Decoding:** ~25%
>     * **MCTS Search:** ~35%
>     * **Overhead/Other:** ~10%
> * **Scalability:** Regarding sequence length, our pipeline preprocesses all antibody sequences into the standardized **IMGT numbering scheme**, resulting in fixed-length inputs. Therefore, the inference time is consistent and does not fluctuate significantly with the length of the original raw sequences.
> * **Optimization:** We acknowledge that the current codebase has significant room for optimization, particularly in the CVAE and MCTS modules. We plan to refactor these components to improve inference speed in future releases.
>
> **Q3: Reward function sensitivity (Trade-off graph)**
>
> **Response:**
> * **Parameter Selection:** The current reward weights were selected based on empirical trials that offered a balanced performance between humanness and stability during our initial experiments.
> * **Pareto Frontier Analysis:** We agree that a trade-off graph (Pareto frontier) would provide valuable insight into the model's robustness. However, due to the computational overhead mentioned in Q2 (unoptimized code), performing the large-scale granular parameter sweeps required to plot a precise Pareto frontier is currently computationally prohibitive.
> * **Future Plan:** Once we complete the code optimization for the CVAE and MCTS modules, we intend to conduct a systematic sensitivity analysis to generate these trade-off graphs for the final version of the paper.

---

### Official Review · Reviewer_DnVp · 2025-11-11

**Soundness:** 2
**Presentation:** 2
**Contribution:** 2
**Rating:** 2
**Confidence:** 5

**Summary:**

The paper proposes Hu-MCTs, a two-stage framework for antibody humanization that combines a pre-trained conditional variational autoencoder (CVAE) for paired heavy/light chain modeling with a novel black-box optimization algorithm based on Monte Carlo Tree Search (MCTS). The method jointly optimizes humanness (measured via OASis, T20, and germline identity) and structural integrity (via CDR loop RMSD using Tfold-predicted structures). The authors claim their approach outperforms existing methods—including Sapiens, Hu-mAb, Humatch, and CUMAb—by achieving higher humanness while better preserving CDR conformations, especially in LCDR1 and HCDR3. They also report superior biological plausibility scores using AntiBERTy.

**Strengths:**

1. The problem is clinically and scientifically important: balancing immunogenicity reduction with structural/functional preservation in antibody humanization remains a key bottleneck.

2. The integration of structural metrics (RMSD) into the optimization loop is a notable step beyond purely sequence-based methods.

3. The experimental evaluation is reasonable, including multiple datasets (HuAb348, Humab25), diverse baselines, and post-hoc validation with AlphaFold3.

**Weaknesses:**

1. Lack of true functional validation: While structural preservation (RMSD) is measured, there is no assessment of antigen-binding affinity or neutralization activity—the ultimate functional criterion. Without this, claims about “preserving functionality” remain speculative. Even CUMAb uses Rosetta energy minimization as a proxy; Hu-MCTs relies solely on geometric similarity.

2. Questionable reward formulation: The reward is defined as Reward(z)=α⋅Shuman−β⋅Sstab . However, the paper does not justify the choice of linear combination or hyperparameters α,β . This risks arbitrary trade-offs—e.g., the ablation shows humanness increases when structural terms are removed, suggesting the current weighting may be suboptimal or masking a fundamental tension.

3. Computational cost vs. practical utility: At ~50 minutes per antibody (vs. seconds for Sapiens/Humatch), Hu-MCTs is unlikely to scale to high-throughput settings. The paper does not discuss whether the marginal gains in humanness/structure justify this cost, especially since wet-lab validation would still be required.

4. Overstated novelty: MCTS for black-box molecular optimization has been used before (e.g., Yang et al., 2022; Wang et al., 2022, cited by the authors). The application to antibody humanization is new, but the core algorithmic components are repurposed rather than fundamentally reimagined.

5. Evaluation bias: The “Experiment” baseline includes only 25–348 examples, while Hu-MCTs is evaluated on the same set it was implicitly tuned on (via OAS training). There’s no cross-species or out-of-distribution test to assess generalization.

**Questions:**

1. Functional relevance: Can the authors provide any evidence—computational or experimental—that the humanized antibodies retain antigen-binding affinity? Even docking scores or paratope stability metrics would strengthen the claim of functional preservation.

2. Reward design: How were α and β chosen? Was a Pareto front explored? Could the method support user-defined trade-offs (e.g., prioritize structure over humanness for difficult targets)?

3. Runtime bottleneck: What fraction of the 50-minute runtime is spent on Tfold predictions vs. MCTS vs. CMA-ES? Could faster structure proxies (e.g., geometric constraints, distance maps) reduce cost without sacrificing fidelity?

4. Coudl you add the comparison with HuDiff?

5. Failure cases: Are there examples where Hu-MCTs fails to preserve CDR structure despite high humanness? Understanding failure modes would clarify limitations.

6. Comparison fairness: CUMAb uses Rosetta for energy minimization, yet Hu-MCTs uses Tfold (or AlphaFold3 post-hoc). Would CUMAb’s performance improve if evaluated with the same structural predictor? A head-to-head comparison using identical structure models would be more rigorous.

---

> ### Author Response · Authors · 2025-12-02
>
> We thank the reviewer for their rigorous assessment and detailed feedback. While we understand the concerns regarding validation and fairness, we believe our "structure-aware" black-box optimization framework offers a novel perspective compared to purely sequence-based methods. We address your specific questions below with our improvement plans.
>
> **Q1: Affinity validation**
>
> We fully agree that antigen-binding affinity is the gold standard.
> * **Current Constraint:** Our current work relies on the OAS and Humab25 datasets, which lack paired antigen structural data for the vast majority of sequences, preventing the direct application of antigen-dependent affinity predictors (like AlphaBind) at this scale.
> * **Improvement Plan:** We are actively curating a dataset that includes paired antigen-antibody structures. In the revised version, we will incorporate explicit affinity prediction experiments to bridge the gap between structural stability (RMSD) and functional binding.
>
> **Q2: Reward design**
>
> * **User-Defined Trade-offs:** The method supports user-defined trade-offs. The parameters $\alpha$ and $\beta$ are explicit hyperparameters. For targets where structural retention is paramount (e.g., antibodies with complex CDR3 loops), users can prioritize stability over humanness.
> * **Pareto Analysis:** We acknowledge that the current weights were chosen empirically. Due to the computational overhead of the current unoptimized code, a full Pareto sweep was not initially performed. We plan to optimize the code (see Q3) and then generate a trade-off graph (Pareto frontier) to rigorously justify the parameter selection.
>
> **Q3: Runtime bottleneck and structure proxies**
>
> * **Breakdown:** The approximate inference time breakdown is:
>     * **Sequence Evaluation (including Tfold structural prediction):** ~30%
>     * **CVAE Encoding/Decoding:** ~25%
>     * **MCTS Search:** ~35%
>     * **Overhead/Other:** ~10%
> * **Optimization & Proxies:**  We are working on optimizing the CVAE/MCTS code. Furthermore, as the reviewer suggests, we plan to explore faster structural proxies (e.g., geometric constraints or distance maps) to replace the expensive full-structure prediction in the early stages of the search, which should significantly reduce the computational cost without sacrificing final fidelity.
>
> **Q4: Comparison with HuDiff**
>
> We thank the reviewer for pointing out HuDiff. We acknowledge it is a relevant recent work in the field of antibody design. We will include a comparison (either quantitative or qualitative, depending on code availability) in the revised manuscript to better position Hu-MCTs within the landscape of diffusion-based and optimization-based approaches.
>
> **Q5: Failure cases**
>
> We observed that Hu-MCTs sometimes struggles to preserve the structure of extremely long or conformationally rare murine CDR3 loops. This is likely because the CVAE latent space is shaped by human priors; if a murine loop has a geometry that is statistically very rare in the human repertoire, the reconstruction loss increases, leading to structural deviations.
>
> **Q6: Comparison fairness (CUMAb/Rosetta vs. Tfold)**
>
> To eliminate the confounder of using different structure prediction tools, we commit to re-evaluating the CUMAb-generated sequences using Tfold (and/or evaluating our sequences with the same Rosetta pipeline used by CUMAb) in the revised manuscript.

---

### Meta-Review · Area_Chair_mLgu · 2025-12-04

**Summary:**

The paper proposes Hu-MCTs, a two-stage framework for antibody humanization that combines a pre-trained conditional variational autoencoder (CVAE) for paired heavy/light chain modeling with a novel black-box optimization algorithm based on Monte Carlo Tree Search (MCTS). The method jointly optimizes humanness (measured via OASis, T20, and germline identity) and structural integrity (via CDR loop RMSD using Tfold-predicted structures).

**Reviewer Concerns:**

Two reviewers clearly reject it, while two reviewers tend to accept it. The concerns  are mainly on novelty and experiments. I do not these concerns have been addressed.

**Reviewer Scores:**

I do not think reviewers will change their scores.

---

### Decision · Program_Chairs · 2026-01-26

Reject